# Unfavorable Outcome and Long-Term Sequelae in Cases with Severe COVID-19

**DOI:** 10.3390/v15020485

**Published:** 2023-02-09

**Authors:** Andrea Fabbri, Antonio Voza, Alessandro Riccardi, Simone Vanni, Fabio De Iaco

**Affiliations:** 1Emergency Department, AUSL Romagna, Presidio Ospedaliero Morgagni-Pierantoni, 47121 Forlì, FC, Italy; 2Emergency Department, Humanitas Research Hospital, 20089 Milan, MI, Italy; 3Unit of Emergency Medicine, Ospedale San Paolo, 17121 Savona, SV, Italy; 4Dipartimento Emergenza e Area Critica, Azienda USL Toscana Centro Struttura Complessa di Medicina d’Urgenza, 50053 Empoli, FI, Italy; 5Struttura Complessa di Medicina di Emergenza Urgenza Ospedale Maria Vittoria, ASL Città di Torino, 10144 Torino, TO, Italy

**Keywords:** outcome, risk factors, long-term complications, post-acute sequelae, prevalence, symptoms

## Abstract

Emerging evidence shows that individuals with COVID-19 who survive the acute phase of illness may experience lingering symptoms in the following months. There is no clear indication as to whether these symptoms persist for a short time before resolving or if they persist for a long time. In this review, we will describe the symptoms that persist over time and possible predictors in the acute phase that indicate long-term persistence. Based on the literature available to date, fatigue/weakness, dyspnea, arthromyalgia, depression, anxiety, memory loss, slowing down, difficulty concentrating and insomnia are the most commonly reported persistent long-term symptoms. The extent and persistence of these in long-term follow-up is not clear as there are still no quality studies available. The evidence available today indicates that female subjects and those with a more severe initial disease are more likely to suffer permanent sequelae one year after the acute phase. To understand these complications, and to experiment with interventions and treatments for those at greater risk, we must first understand the physio-pathological mechanisms that sustain them.

## 1. Introduction

The SARS-CoV-2 disease (COVID-19) has been found to have an unprecedented impact on all aspects of human activity worldwide [1]. Given the impact of a widespread vaccination campaign, the results of antiviral drugs and monoclonal antibodies, uncertainties remain regarding the predictors of both short- and long-term prognosis of complications in the acute phase and long-term sequelae.

Given the public health implications of these factors, a better understanding of the pathophysiology of the disease could help predict the short- and long-term prognosis.

The evolution of the disease in terms of clinical features suggests an increasing number of additional clinical factors associated with patient characteristics. Predictors of disease progression include demographic, clinical, immunologic, hematologic, biochemical and radiologic findings that may be useful to physicians in predicting the severity and course of the disease.

## 2. Unfavorable Outcome

The clinical presentation is usually a respiratory infection with symptom severity ranging from a mild common cold-like illness to that of severe viral pneumonia complicated by acute respiratory distress syndrome and multiple organ failure.

### 2.1. Mortality

The most common cause of death in COVID-19 patients is respiratory failure due to acute respiratory distress syndrome (ARDS) [2]. The overall ARDS mortality rate in COVID-19 patients is more than 1/3 of cases; however, data vary and are very heterogeneous in the different countries considered (China 69%, Iran 28%, France 19%, Germany 13%) [3].

An insufficient correlation is reported between poor prognosis (e.g., days without mechanical ventilation, length of stay in ICU or hospital, or mortality) in cases with COVID-19 ARDS and cases with ARDS but without COVID-19 [4]. The most important risk factors for respiratory failure are proved advanced age, male sex, concomitant diseases, especially cardiovascular disease, laboratory markers (such as lactate dehydrogenase, lymphocyte count and C-reactive protein) and a high viral load on admission [5,6].

Many risk factors for mortality include comorbidities [7,8,9,10], such as diabetes mellitus, obesity, systemic hypertension, renal disease, coronary heart disease [11], malignancy [12,13] and elevated levels of creatinine [8], lactate dehydrogenase [11,12,14,15], direct bilirubin [12] and alanine aminotransferase [8], which provide early indications of disease severity; elevated plasma levels of biomarkers such as D-dimers [9,12], C-reactive protein [10,16], serum ferritin [8] and procalcitonin further contribute to increased risk [8,9,14].

Mortality results are high among critically ill patients: overall mortality is 35% in intensive care units, a percentage not different from the 32% mortality in all critically ill patients admitted to hospital. These percentages may vary greatly from region to region (48% in Southeast Asia versus 15% in the US [17]) and within the same study (32.3% vs. 16.4% in a UK registry [18]).

Mortality rates also declined during in the first 6 months of the pandemic period [19], namely in cases with severe disease admitted to an intensive care unit (43.5% vs. 19.2% [20]. This trend could be the result of changes in specific hospital policies and clinical procedures, as well as improved adherence to evidence-based standard therapies for critical cases, such as the use of corticosteroids, high-flow nasal oxygen, prone positioning and reduced use of mechanical ventilation [20]. In the Emergency Department setting, a group of early variables (age, sex, number of concomitant diseases, respiratory rate, peripheral oxygen saturation ≥ 92% in the room, Glasgow Coma Scale, plasma urea and C-reactive protein) proved a good discriminatory power in the prediction of unfavorable outcomes [21].

Although several prognostic models have been proposed over the years and some have been accepted by the scientific community, the predictive value of most of these models has not yet been definitively validated in large clinical trials.

Due to the ever-changing viral dynamics, more and more new symptoms have been observed and recorded. It is hoped that as new data continue to be added, the predictive models will gradually refine the pattern of symptoms and the characteristics of the major clinical manifestations, as well as the group of laboratory data, radiological examinations and therapies that are useful to better characterize the different degrees of disease.

### 2.2. Co-Infections (Bacterial, Fungal and Parasitic Infections)

In the acute phase of the SARS-CoV-2 disease, bacterial infections represent an important complication [22]. A confirmed bacterial infection was reported in 8.0% (95% confidence intervals (95%CI) from 6.1% to 9.9%) of cases, with a higher prevalence of secondary infections (16.0%, 95%CI from 12.4% to 19.6%) compared with co-infections (4.9%, 95%CI from 2.6% to 7.1%). Although this analysis included a total of more than 6000 cases, it should be considered that most of the studies were retrospective, of low quality and of small dimension; in fact, as many as 13 studies considered a number of cases lower than 50 and, moreover, data on the timing of infection, microbiological details and antimicrobial use were insufficient.

The prevalence of a bacterial co-infection in subjects with acute SARS-CoV-2 disease was 12%, but with 95%CI from 8% to 16% [23]; this range is dependent on the initial disease severity. In the general population of the studies considered, co-infections were registered in an average of 4% (95%CI from 3 to 6%) of cases, but in studies on critically ill patients, it reached 12% (95%CI from 4% to 22%) [22].

This prevalence was reduced when diagnosis was obtained at presentation: 3.5% (95% CI, 0.4 to 6.7%), but it was greater as a complication acquired during hospitalization in 14.3% (CI 95% from 9.6% to 18.9%) of cases [22].

In a recent review analysis [24], up to 35% of cases were co-infected with *Enterobacter* spp., 27% with methicillin-susceptible Staphylococcus aureus, 21% with *Klebsiella* spp. 6% coagulase-negative Staphylococcus, 13% with *Escherichia coli* and 3% with *Pseudomonas aeruginosa*. The finding of these co-infections was documented in the blood stream, while in the urinary and respiratory tract, *Streptococcus pneumoniae* was found in 57%, Staphylococcus coagulase negative in 44% and *Escherichia coli* in 37% of cases [24].

Due to the implications both in terms of health and management, reinfection after first episode of COVID-19 remains an issue still under discussion [25]. The dynamics of the virus and its mutation in the most important countries lead to resistance in therapy and vaccination [26]. To date, available data suggest that the reinfection rate in several countries may be between 0.5% and 5% of cases [27].

Reinfection seems to lead to an approximately 2-fold increase in mortality, an approximately 3-fold increase in hospitalization, and a significant increase in pulmonary, cardiovascular, hematological, gastrointestinal, renal, musculo-skeletal and neurological complications [28]. The results indicate that the risk of short- and long-term complications of post-acute syndrome at 6 months is increased in relation to the number of infections, irrespectively of preventive vaccination [28].

## 3. Long Term Sequelae

The epidemiological and clinical features as well as the acute complications of patients with acute COVID-19 have been described in detail [29], but the long-term implications of this disease are still largely unclear. So far, only studies with a follow-up of 3 months after discharge have been published [30,31], and they are limited to persistence or clinical aspects such as physical and psychological aspects [32]. They refer to persistent symptoms such as fatigue, dyspnea [33,34], pulmonary insufficiency [31,35] and abnormalities in the chest radiograph after hospital discharge [30], but the full range of post-discharge characteristics is still unknown.

Post-discharge health outcomes have been reported following acute illness, and up to 80% of patients may continue to complain of problems after the acute phase, with over 50 categories of adverse events reported [36]. The pathophysiology of many of these post-acute symptoms remains unclear, and symptoms can sometimes persist for several weeks, delay recovery or even recur over 3 months after the acute phase.

### 3.1. Pathophysiology

The SARS-CoV-2 disease is characterized by a potent primary viral neutralizing immune response, and a subsequent specific immunological response [37] to severe disease. The uncontrolled replication of the virus evades the primary immune activation by the host by stimulating an important inflammatory response with the recruitment of inflammatory monocytes, neutrophils, activated T cells and consequent tissue damage, particularly in the lung [38].

An abnormal inflammatory response to high levels of potent inflammatory cytokines, particularly tumor necrosis factor and interleukine-6, further activates downstream immunopathological events, including the activation of the coagulation cascade and subsequent multiorgan damage [39]. To date, the immunological link between the acute phase of the disease and the development of a post-COVID-19 syndrome is still debated.

Being able to distinguish between persistent symptoms in the acute phase of the disease and those that are added in the later phases of the disease could help in a more reliable diagnosis. There are, for example, disorders attributable to a prolongation of the initial symptom picture that can be included among those of a post-traumatic stress syndrome, or those in the neuropsychiatric field.

Several hypotheses could explain the pathophysiology behind the persistence of symptoms [40]. For example, the degree of inflammatory response to a new infectious agent could delay the complete resolution of the inflammatory picture and, therefore, the persistence of respiratory symptoms in cases with severe lung disease.

In a prospective study [41] on 31 subjects with long-term COVID-19 (diagnosis based on finding of at least one of three main symptoms: dyspnea, fatigue and chest pain) and a control group of asymptomatic subjects with only a history of illness, an 8-month comparative analysis showed that patients with long COVID-19 had persistent increases in activated lymphocytes CD14+, CD16+, monocytes, plasmacytoid, dendritic cells and type I (IFNβ) and type III (IFNλ1) interferon levels [41].

The diagnosis of long COVID-19 was, therefore, associated with an increase in the levels of IFNβ, pentraxin 3, IFNγ, IFNλ2/3 and IL-6, with an accuracy ranging from 78.5% to 81.6% [41]. These findings indicate that subjects with persistent symptoms and an increased inflammatory mediator pattern were those with some sort of “delayed” resolution of the inflammatory condition. At the other end, there are cases where a “hyperinflammatory” response can leads to a condition of pulmonary fibrosis and, thus, a high likelihood of impaired respiratory function [41].

Although certain symptoms such as fatigue and dyspnea are not often associated with the level of inflammation, increases in some markers of persistent neutrophil activation (e.g., lipocalin-2), fibrosis signaling (matrix metalloprotease-7) and cell repair alveolar epithelium (hepatocyte growth factor) indicate a direct relationship with the severity of the inflammation and, therefore, of respiratory function deficit [42].

A recent hypothesis suggests that the SARS-CoV-2 virus could modify the mechanisms of immune homeostasis by regulating tissue inflammation by determining persistent lung lesions [43]. This persistent increase in some inflammatory mediators, namely IL-6, IL-1β and TNF, could induce complications at the single organ level, in particular, pathologic cardiac “remodeling”, cardiac arrhythmias [44], neuroinflammatory syndromes [45], neuro-degenerative diseases [46], renal failure [47] and peripheral insulin resistance [48].

Other markers, such as the persistent increase in T and B effector cells, the activation of oxidative phosphorylation, reactive oxygen species and heme-related metabolic pathways, can be implicated as a cause of the delayed “immunological recovery” and, therefore, of long-term sequelae [37].

In conclusion, a great deal remains to be understood regarding the persistence of SARS-CoV-2 in humans. It is not clear whether the persistence of the virus in some systems, the intestine, for example, favors or hinders its long-term survival, which may influence the long-lasting symptoms of COVID-19.

### 3.2. Vaccines and Long COVID-19

Preventative vaccination for acute COVID-19 is expected to reduce not only the incidence of the disease, but also its long-term effects [49]. The studies published so far are not conclusive, and current opinions question whether vaccines may have a role in the development of symptoms of COVID-19 in the long term [50,51,52].

The first question is whether vaccines prevent or do not prevent the development of COVID-19 in the long term. The six studies available to date (methodological quality ranging from moderate to high) conclude that the preventive vaccine reduces the risk of developing long-term COVID-19, but only in cases with a mild to moderate acute disease. However, it should be considered that most of the studies evaluated a “short-term” effect of the vaccine, as most involved a history of disease between one week and one month after vaccination, and only two studies considered a follow-up 6 months after being vaccinated [53,54].

The definition of long COVID-19 is very different between studies, and non-definitive data suggest that two doses of the vaccine are more effective than a single dose, and that the COV2.S vaccine (“Janssen”) [54] is able to reduce the risk of long COVID-19. Furthermore, no studies are available on the subject of vaccination recalls and long-term COVID-19, nor are any physio-pathological mechanisms that would support their effect being known.

Two hypotheses were formulated: (1) since vaccines reduce the severity of acute infection, this can consequently reduce the risk of short- and long-term complications, even if the association of long-term COVID-19 with disease severity remains controversial [55], and (2) vaccines accelerate the clearance of the virus (see the association between viral residue and long COVID-19), and consequently reduce the associated exaggerated inflammatory and/or immune response (see the immune/inflammatory theory of long COVID-19) [56].

A further aspect to be considered is whether COVID-19 vaccines are a risk of complication in those subjects with long-standing COVID-19 symptoms. In the 11 studies (level III, moderate to high methodological quality, wide-definition heterogeneity) on individuals with a history of acute illness and with long COVID-19 (long-infection COVID-19 vaccine design), the results are not conclusive, as benefits were reported in only 2/3 of the studies; meanwhile, in more than 1/3 of the studies, there were no significant benefits or even a worsening in the clinical picture [49]. In the absence of strong evidence, we can speculate that vaccination may help reduce long-term COVID-19 by eliminating the viral reservoir or restoring a dysregulated immune response to acute primary infection. This effect could vary with the host.

## 4. Post-COVID-19 Syndrome

While long COVID-19 is defined as when symptoms persist for more than 4 weeks after the acute phase, post-COVID-19 syndrome is now considered a clinical condition in which symptoms persist for at least 2 months after the onset of acute illness or at least 3 months after acute infection, unless another diagnosis is made [57].

In a retrospective study, the prevalence of SARS-CoV-2 post-acute sequelae was 11%, or 3 out of 4 asymptomatic patients with a 30-day follow-up. Five categories of specific symptoms were highlighted on days 0–30, representative of post-acute sequelae from SARS-CoV-2 [58]. The model identified 27% of cases with symptoms predictive of long-term disease. Women were more likely to work as long haulers than men, and all age groups were represented, with those aged 50 to 20 years accounting for 72% of cases. The symptoms following the 60-day follow-up identifying post-COVID-19 syndrome were: chest pain–cough, dyspnea–cough, anxiety–tachycardia, abdominal pain–nausea, and lumbar–joint pain [58].

To date, no definitive studies on the prevalence of the disease and the characteristics of post-COVID-19 six months after acute infection have been published. Unfortunately, data on quality of life and work ability are also lacking.

In a recent study [35], 76% of patients reported at least one symptom 6 months after the onset of symptoms, with a higher proportion in women. The most common symptoms were muscle fatigue or weakness and sleep problems. In addition, 23% of patients reported anxiety and depression at follow-up. The percentage of patients with a pulmonary diffusion abnormality at follow-up is higher in patients with more severe disease in the acute phase [35].

Cases with suspected post-COVID-19 sequelae have a variety of symptoms affecting different groups and combinations of organs. These symptoms cannot be attributed to any other cause, as the only significant event was recent COVID-19. Studies in this area are scarce: almost 90% of COVID-19 survivors developed sequelae, including not only general symptoms such as fatigue, but also severe neurological, cardiac, renal or respiratory manifestations [40]. In some cases, symptoms may persist and last longer than 12 weeks, especially if patients had severe ARDS and multi-organ failure.

Persistent sequelae of COVID-19 are, however, thought to be common in people with the following risk factors: older people (over 50 years of age), smokers and people with comorbidities such as hypertension, obesity, diabetes, chronic lung disease, cardiovascular disease, chronic kidney disease, chronic liver disease, cerebrovascular disease, cancer and immunodeficiency [40].

Following milder SARS-CoV-2 infections, infectious sequelae have been observed in observational studies collecting prescription data or electronic health records at 6 months follow-up [59,60,61]. Persistent dyspnea is often associated with lung damage and impaired lung function in postmortem lung tissue [62]. Furthermore, fatigue as part of COVID-19 sequelae does not appear to be associated with dysfunction, although SARS-CoV-2 has also been detected in endothelial cells [40]. SARS-CoV-2 particles were also detected under electron microscopy in penile tissue samples, suggesting a link between COVID-19 sequelae and erectile dysfunction [63]. Consistent with the observed vascular damage, endothelial dysfunction has been reported after recovery from COVID-19 as the gold standard method (i.e., flow-mediated dilation).

Previous SARS-CoV-2 infection was an independent predictor of impaired flow-mediated dilation [63]. Increased inflammatory responses, oxidative stress, proinflammatory cytokines and reduced mitochondrial function have also been described in the pathophysiology of COVID-19 sequelae [63]. The pattern of morphological and functional changes at the endothelial level may be the main sequelae, which have been characterized as long-lasting or post-COVID-19 [64]. Studies in this area are scarce: about 90% of survivors reported sequelae, including not only general symptoms such as fatigue, but also severe neurological, cardiac, renal and respiratory manifestations [65].

According to a UK Biobank study, COVID-19 has also been associated with long-term changes in brain structure [66]. In a telephone survey of adults who tested positive for SARS-CoV-2, 95 (35%) of the 274 symptomatic respondents, including 22 (26%) of the 85 subjects aged 18–34 years, reported that they had not returned to their usual state of health two weeks or more after testing.

The Pan American Health Organization has issued an epidemiological alert on the need for information on this issue. The World Health Organization (WHO) has added the term “post COVID-19 condition” to the International Classification of Diseases codes to describe a condition that occurs as a result of probable or confirmed SARS-CoV-2 infection, whose symptoms last for at least two months, and which cannot be explained by any other diagnosis [67].

We will now review the main studies that support what is currently known on this topic. In a recent study [68], multi-organ function was assessed by a series of sensitive and minimally invasive tests in a group of subjects several months after COVID-19. The interesting thing about the study was the control group: the control group was shown by a confirmatory serological test to have had no exposure to SARS-CoV-2, unlike other studies that looked at general health. The study [68] focused on organ function, i.e., cardiovascular function, pulmonary function, neurological function, ophthalmological function, male fertility and psychological aspects, using a battery of innovative tests in a clinical setting rather than online or using telephone surveys or database analysis. The study also adds to the evidence on the spectrum and persistence of COVID-19 sequelae in previously healthy young adults. The findings presented here demonstrated a delay in full recovery of the general condition without long-term consequences after COVID-19, albeit to a minor extent (defined as symptomatic but not hospitalized) up to 180 days after illness.

Persistent sequelae included an increase in body mass index, the finding of dyslipidemia and a decrease in physical performance over many months post-infection with a reduction in aerobic threshold. It was also found that the cases of hyposmia and a deficit in active sperm count were not related to the duration of follow-up (180 days). The anxiety and depression status scales showed greater psychological consequences in the short-term than the long-term follow-up (180 days) [68].

Overall, 10 months after infection, complete recovery from sequelae and the disappearance of most complications are achieved, but some metabolic complications may still occur. Only young patients who were previously considered healthy and did not require hospitalization recovered fully after a mild infection, as multisystem involvement of the disease is less common than in older or hospitalized individuals. However, the conclusions [68] provide new evidence that even mild infections in young adults can lead to sequelae that persist for several months after the infection has healed.

The most common sequelae after COVID-19 were observed in older women, especially in those who suffered severe complications during the acute phase of infection [69,70]. More than 200 sequelae have been reported, the most common being breathing difficulties, changes in taste and smell, fatigue and neuropsychological symptoms such as memory loss, anxiety and depression [65,70]. Some of the reported symptoms have been shown to persist until the fifth week in 20% of patients and until the twelfth week in 10% of patients, especially in women aged 36–50 years with previous comorbidities [69]. Associated risk factors also include female sex, more than five early symptoms in the acute phase of COVID-19, early dyspnea, previous psychiatric disorders and an altered D-dimer, C-reactive protein and lymphocyte count [69,70].

An additional study reported that only 65% of referred patients recovered their pre-infection health status within 2–3 weeks [71]. Although some COVID-19 sequelae are usually resolved spontaneously, they can still be very disabling, leading to prolonged bed rest and additional restrictions on daily activities, which impair physical performance and increase fatigue and breathlessness [72,73]. In these cases, the increase in bronchial secretions, the appearance of fibrosis and the weakness of the respiratory muscles lead to an impaired ventilatory pattern [74], which may affect the patient’s work activities, family responsibilities and quality of life [69,73].

Persistent symptoms after the acute stage of the disease can occur in a very high frequency. According to a recent review [75], more than 72% of cases had at least one symptom more than 60 days after initial diagnosis or symptom onset, or 30 days after hospitalization or discharge from hospital. These persistent symptoms were most often reported as especially debilitating conditions such as tiredness and shortness of breath. On the other hand, the persistence of atypical chest pain was reported in about 1 in 7 patients, as was the inability to concentrate, also called “brain fog,”, in 1 in 4 patients, although this symptom was reported in only four studies, whereas other neuro-cognitive deficits were represented in an equal percentage (Figure 1).

These results, combined with images documenting the damage to the affected organs, lead to a pathophysiological justification of the reported symptoms. For example, positive imaging data for brain damage, including those of the brain regions responsible for smell and memory, are more represented in subjects with COVID-19 than in healthy subjects, as well as those with myocardial damage [75]. These findings, combined with imaging documentation of the damage to the affected organs, result in the pathophysiological justification for the reported symptoms. 

## 5. Conclusions

In subjects with a diagnosis of COVID-19, identifying the key factors that predict with sufficient accuracy the likelihood of disease exacerbation is very important. Several predictors, including advanced age, the patient’s risk profile, the number of comorbidities, the nature of the immune response, aspects of diagnostic imaging, some laboratory markers and indicators of organ dysfunction, all may predict a poor outcome.

Advanced age, the patient’s risk profile, number of comorbidities, the type of immune response, features of diagnostic imaging, various laboratory markers and symptoms of organ failure are just a few predictors that may indicate a poor outcome.

The difficulty in predicting the severity of the COVID-19 disease with sufficient accuracy is underscored by the fact that SARS-CoV-2 appears to exhibit tropism for several tissues, including primarily the respiratory tract, but also the brain, endothelium, heart, kidneys and liver. The increasingly accurate identification of all these elements could be of great help in guiding clinical care, improving patient outcomes and deploying appropriate resources.

Post-COVID-19 syndrome includes a wide range of symptoms with potential long-term consequences 6–12 months after SARS-CoV-2 infection, affecting both general health and the ability to work. The pattern of morphological and functional changes at the level of the endothelia could be the main consequence, which can be described as long-lasting or post-COVID-19. So far, however, the pathophysiology has not been fully clarified.

Although a variety of long-lasting symptoms have been reported, only a few are actually associated with post-COVID-19 syndrome: fatigue, neurocognitive deterioration, shortness of breath, impaired muscle strength and a reduced quality of life. The occurrence of these symptoms, which are more frequent in female subjects and in cases with more severe disease, negatively affect both the recovery of the disease and the ability to work. Bacterial, fungal or parasitic co-infections are considered an important complication for the prognosis, but available studies are low-quality and heterogeneous.

The risk of long-term complications is likely to increase in relation to the number of acute SARS-CoV-2 infections, but no significant relationship between long COVID-19 syndrome and vaccination is proven. There is little evidence to support the effects of the vaccine in reducing the risk of long COVID-19 syndrome.

## Figures and Tables

**Figure 1 viruses-15-00485-f001:**
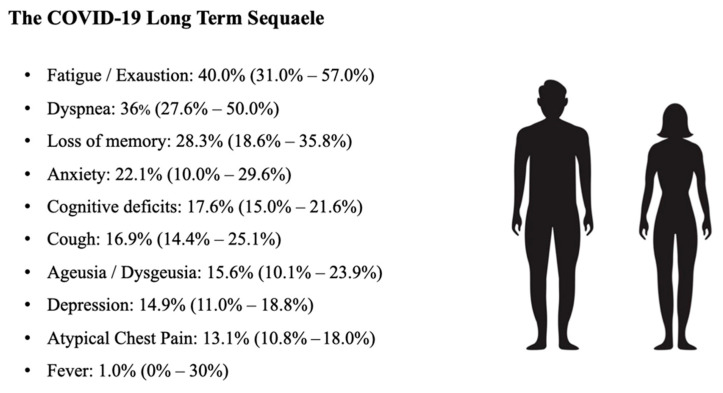
Reported frequencies of the 10 most reported long-term sequelae in patients with COVID-19, from a systematic review analysis. Data are reported in order of frequency as median and interquartile range [75].

## Data Availability

Not applicable.

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
