# Peer review of "Unfavorable Outcome and Long-Term Sequelae in Cases with Severe COVID-19"

_viruses, 2023, doi:10.3390/v15020485_

Round 1

Reviewer 1 Report

The topic of long COVID is an important and difficult one, and the authors attempt to summarize the available data on the topic. 

1. Overall though well written, much of this paper still focuses on acute COVID outcomes/symptoms rather than long COVID. Section 2 is basically a general review of COVID, and it's not clear how it relates to the goal of evaluating long COVID, which starts in section 3. 

2. Page 4, Line 198: "In late 2020, a lengthy forum (COVID).. "  The forum was called COVID?

3. Even in section 3, the paper jumps back and forth between early and late symptoms and between being chronological between publication date and symptom based. Would suggest some reorganization to help the flow.. ie Discuss the available data for each symptom thought to be related to long COVID or discuss chronologically what we known and built upon. 

4. Also not addressed is the impact of multiple COVID infections and the impact of vaccines on long COVID symptoms. Would suggest removing section 2 and adding the available data on these areas to make the summary more robust. 

5. Maybe a table with the different studies showing percentages of each symptom to help summarize. 

6. The conclusion talks about predictive factors, but very little is said above about those. So similarly maybe a section or a table of all the different predictive factors from the different studies..  and then in the conclusion, the authors can suggest from their review which have the most evidence to study going forward.

Author Response

I appreciate the Reviewers' thorough criticisms and advice, which I will take into consideration. They'll undoubtedly make the actual version better. As the Editor advised, the text has been extended to include more than 4000 words (4254), which has resulted in a noticeably higher number of references.

Point by point response to Rev comments:

Reviewer #1

  1. The topic of long COVID is an important and difficult one, and the authors attempt to summarize the available data on the topic. Overall, though well written, much of this paper still focuses on acute COVID outcomes/symptoms rather than long COVID. Section 2 is basically a general review of COVID, and it's not clear how it relates to the goal of evaluating long COVID, which starts in section 3: Replay: Replay: Paragraph 2.1 on mortality and Paragraph 2.2 on co-infections were added to Section 2, which had been renamed from Prognosis to Unfavorable result. A section on multiple COVID infections was also added to the para's final section.)

  1. Page 4, Line 198: "In late 2020, a lengthy forum (COVID).. " The forum was called COVID?:

Replay: corrected.

  1. Even in section 3, the paper jumps back and forth between early and late symptoms and between being chronological between publication date and symptom based. Would suggest some reorganization to help the flow.. i.e. Discuss the available data for each symptom thought to be related to long COVID or discuss chronologically what we known and built upon.

Replay: the section was re-written according to the rev suggestions.

  1. Also not addressed is the impact of multiple COVID infections and the impact of vaccines on long COVID symptoms. Would suggest removing section 2 and adding the available data on these areas to make the summary more robust:

Replay: In the final section of section 2.2, a paragraph on repeated COVID infections was added, and a specific paragraph on the connection between vaccines and the lengthy COVID syndrome was added..

  1. Maybe a table with the different studies showing percentages of each symptom to help summarize: Replay: Figure 1 was amended according to the Rev. suggestion. The frequencies of 10 most reported long-term sequelae were indicated (median and interquartile range) as reported in the systematic review of Nasserie T, et al. Assessment of the Frequency and Variety of Persistent Symptoms Among Patients With COVID-19: A Systematic Review. JAMA Netw Open. 2021 [Ref 76].

  1. The conclusion talks about predictive factors, but very little is said above about those. So similarly maybe a section or a table of all the different predictive factors from the different studies .. and then in the conclusion, the authors can suggest from their review which have the most evidence to study going forward. Replay: Conclusions were rewritten according to rev suggestions.

Reviewer 2 Report

This manuscript entitled “Unfavorable Outcome and Long-term Sequelae in Cases 2 with Severe COVID-19” by Andrea Fabbri et al. presents recently published reports over COVID-19 sequelae in a consolidated review form for a better understanding on predicting the severity of long term COVID-19 effects. The review article is well written and structured but have some grammatical errors like missing punctuations, periods etc. The authors have citied mostly cited recent research articles but adding/discussing few more published reports will enhance the quality of this manuscript. Authors have highlighted majority of the possible COVID-19 sequelae but adding few more to the list would broaden the scope of this manuscript. Authors may look up more research papers such as by Saurabh Mehandru (https://www.nature.com/articles/s41590-021-01104-y) which has discussed possible sequelae and their molecular dynamics; and may also look up healthcare organizations’ medicine websites such as Yale medicine site (https://www.yalemedicine.org/conditions/long-covid-post-acute-sequelae-of-sars-cov-2-infection-pasc). Additionally, I suggest authors to include and discuss the Yong Huang work (https://pubmed.ncbi.nlm.nih.gov/33688670/) which would bring depth and clarity to this report. I suggest authors to include a section of COVID-19 and its association with bacterial, fungal and parasitic infections and how these infectious agents become prominent post COVID-19 infection and discuss possible outcomes.

Author Response

Reviewer #2

General Comments: this manuscript entitled “Unfavorable Outcome ……. Replay: The manuscript was modified in response to Rev recommendations; in particular, Section 2's prognosis was changed to an adverse outcome, and the first paragraph 2.1's discussion on mortality was greatly condensed. In response to the Rev Suggestions, we add paragraph 2.2 on Co-infections and cover the subject of numerous COVID infections in the final paragraph of paragraph 2.2. As a consequence, the reference's number has been significantly increased in accordance with the Rev's recommendations.

Specific Comments:

  1. More research papers by Saurabh Mehandru on possible sequelae and their molecular dynamics;

Replay: the paper by Mehandru was added and largely discussed. To specifically discuss the molecular dynamics of possible long-term sequelae a specific para 3.1 (Page 4 and Page 56) entitled Pathophysiology was added with the aim to report and discuss the molecular causes and implications of long term COVID sequelae.

  1. Yong Huang work: Replay: the paper by Huang on “Symptom Clusters, and Predictors for Becoming a Long-Hauler Looking for Clarity in the Haze of the Pandemic” was added and discussed in Page 5 line 607-614 as suggested by the Rev.
  2. A section of COVID-19 and its association with bacterial, fungal and parasitic infections: Replay: in the section 2, para 2.2 (Page 3 Line 373-407) a specific paragraph on this topic was added and discussed.